# Critical Signaling Transduction Pathways and Intestinal Barrier: Implications for Pathophysiology and Therapeutics

**DOI:** 10.3390/ph16091216

**Published:** 2023-08-29

**Authors:** Jingwang Gao, Bo Cao, Ruiyang Zhao, Hanghang Li, Qixuan Xu, Bo Wei

**Affiliations:** 1Department of General Surgery, Medical School of Chinese PLA, Beijing 100853, China; g18889188977@163.com (J.G.); 15699159786@163.com (R.Z.); lihanghangaaa@163.com (H.L.); xuqixuan0115@163.com (Q.X.); 2Department of General Surgery, First Medical Center, Chinese PLA General Hospital, Beijing 100853, China; caobo_dark@163.com

**Keywords:** intestinal barrier, signaling pathway, microbiota symbiosis, traditional Chinese medicine, clinical translation

## Abstract

The intestinal barrier is a sum of the functions and structures consisting of the intestinal mucosal epithelium, mucus, intestinal flora, secretory immunoglobulins, and digestive juices. It is the first-line defense mechanism that resists nonspecific infections with powerful functions that include physical, endocrine, and immune defenses. Health and physiological homeostasis are greatly dependent on the sturdiness of the intestinal barrier shield, whose dysfunction can contribute to the progression of numerous types of intestinal diseases. Disorders of internal homeostasis may also induce barrier impairment and form vicious cycles during the response to diseases. Therefore, the identification of the underlying mechanisms involved in intestinal barrier function and the development of effective drugs targeting its damage have become popular research topics. Evidence has shown that multiple signaling pathways and corresponding critical molecules are extensively involved in the regulation of the barrier pathophysiological state. Ectopic expression or activation of signaling pathways plays an essential role in the process of shield destruction. Although some drugs, such as molecular or signaling inhibitors, are currently used for the treatment of intestinal diseases, their efficacy cannot meet current medical requirements. In this review, we summarize the current achievements in research on the relationships between the intestinal barrier and signaling pathways. The limitations and future perspectives are also discussed to provide new horizons for targeted therapies for restoring intestinal barrier function that have translational potential.

## 1. Introduction

The intestinal barrier is a unique type of mucosal barrier that can both digest and absorb nutrients and defend against harmful substances such as gut toxic substances, pathogens, and antigens [1]. The intestinal barrier consists of four components: mechanical, immune, chemical, and biological barriers. Each of these components has a corresponding structural basis, together with regulation of the integrity of the intestinal barrier. An integral intestinal barrier plays a vital role in maintaining human health and treating gastrointestinal and even extraintestinal diseases [2]. Many factors related to intestinal barrier damage have been reported, such as ischemia-reperfusion (I/R) injury [3], mental disorders [4], inflammatory mediators [5], microbiota dysbiosis [6], an unhealthy diet [7], bacterial-derived endotoxin [8], immune abnormalities [9], and ionizing radiation [10]. Such factors synergistically trigger the deterioration of organic lesions, systemic inflammatory responses (SIR), and metabolic disorders, causing vicious cycles of homeostasis dysfunction.

Because of the importance of intestinal barrier integrity, the progression mechanisms of the barrier have been deeply investigated. Normally, the intestinal barrier only allows digestive products and water to cross the barrier. However, with continuous exposure to multiple risk factors, the intestinal barrier is inevitably disrupted. Following disruption of the intestinal barrier, intestinal permeability increases. At this time, pathogenic bacteria and their products may enter the lamina propria, causing an immune response and a harmful inflammatory reaction. Subsequently, several inflammatory factors and oxidative stress affect the expression of tight junction proteins, which further increases paracellular permeability [11,12,13]. These pathological changes associated with intestinal barrier disruption interact with each other in a vicious cycle. This makes the treatment of intestinal diseases such as inflammatory bowel disease (IBD), radiation enteritis (RE), and chemotherapy enteritis more difficult [14,15]. How to interrupt pathological changes and restore the function of the intestinal barrier is an urgent issue for many digestive tract diseases.

Signal transduction pathways refer to a series of response pathways that can transmit information molecules from the extracellular space to the intracellular area through the cell membrane or intracellular receptors. These pathways are the basis of all cell biological functions, such as proliferation, differentiation, metabolism, and immune responses [16,17]. Dysregulation of cell signaling pathways is commonly associated with a wide variety of human diseases. The disruption and repair of the intestinal barrier are also dependent on complicated crosstalk between signaling pathways. For instance, Akt acts as a principal switch in the cellular signaling pathway, generating a large number of intracellular responses and interacting with upstream and downstream targets to maintain the integrity of the intestinal mucosa [18,19,20]. The NF-κB pathway is involved in innate and adaptive immunity, the stress response, and lymphoid organogenesis. It is extensively involved in acute and chronic inflammatory responses following intestinal barrier disruption [21]. In summary, the activity of signaling pathways has great significance for the improvement of intestinal homeostasis. Exploration of the mechanisms underlying intestinal barrier repair and the development of related drugs to reverse ectopic signaling pathways may be an effective approach to therapies against intestinal disorders.

## 2. Components of the Intestinal Barrier

### 2.1. Intestinal Epithelium

The intestinal epithelium barrier is a monocellular layer that consists of the intestinal epithelial cells (IECs) of the intestinal mucosa and the basilar membranes. The cell types include enterocytes, endocrine cells, Paneth cells, goblet cells, microfold cells (M cells), and undifferentiated intestinal stem cells (ISCs) [22,23]. Their functions are distinct but collectively contribute to the stability and normal functions of the intestinal barrier. The abundance of enterocytes is the highest. They are responsible for nutrient absorption and barrier integrity. ISCs can continuously proliferate, differentiate, and migrate upward to replenish enterocytes and goblet cells that fall off of the epithelium [24]. The following types of cells with secretory functions are often referred to as secretory cells. Goblet cells pertain to mucus-secreting cells, the numbers of which increase from the duodenum to the anus [25]. Hyperglycosylated mucin is the main secretion product of goblet cells. It is secreted into the enteric cavity and binds with water to produce mucus for intestinal lubrication and saturation [26]. Paneth cells are located in the crypts of the small intestine and make up a small proportion of intestinal epithelial cells [27]. Paneth cells can secrete lysozyme and antimicrobial peptides (AMPs) to regulate bacterial symbiosis [28]. Despite the fact that the proportion of endocrine cells in the intestine is only 1%, the intestine has been thought to be the largest endocrine organ [29,30]. Diverse types of endocrine cells can produce hormones and other signaling molecules to modulate the intestinal environment [31].

M cells and tuft cells are important components of the defense line across the intestinal contents and the host immune system. M cells are phagocytic intestinal epithelial cells in the follicle-associated epithelium of Peyer’s patches and are located in the epithelial zone covered by lymphoid follicles of gut-associated lymphoid tissues. M cells have a significant deficiency of microvilli and the capability of secreting digestive enzymes and mucus. However, the glycocalyx, produced by M cells, facilitates traversing to the enteric cavity and antigen endocytosis [32]. Antigens are transmitted to macrophages and lymphocytes in the epithelium, and immune responses are subsequently induced [33]. Tuft cells are a type of cell with secretory functions found in both the small intestine and colon, but they are very rare [34,35]. This cell type plays an important role in defense against parasites and other pathogenic microorganisms with powerful immunomodulatory potential [36,37]. These IECs, which have absorptive, secretory, and immunomodulatory functions, together form the intestinal epithelial barrier. Disruption of any of these cells could potentially lead to the development of disease.

### 2.2. Tight Junctions

In addition to IECs, tight junctions (TJs), a type of cell junction, are also important components of the mechanical barrier. The TJ is a highly dynamic structure formed by the TJ protein complex and is located on the outer side of the intercellular space of endothelial cells. TJ includes a variety of proteins, such as occluding, claudin, junctional adhesion molecules (JAMs), and zonula occludens (ZO). Claudin and occludin are transmembrane proteins, and ZO is an adaptor protein in the cytoplasm. Their direct binding is involved in the regulation of intestinal mucosal barrier integrity [38,39,40]. TJ plays an important role in sealing the intercellular space, regulating material transport, and maintaining cell polarity [41]. They can even engage in cell migration and proliferation [42].

TJ integrity can prevent the permeation of external antigens and the leakage of internal components. The diffusion of ions, water, and other solutes is also under strict control by TJ [43]. TJ structural and functional disruption is the main cause of increased intestinal permeability. Increased permeability of the epithelial barrier would increase the invasion of harmful substances and thus induce or augment mucosal inflammation. Inflammatory cytokines further reduce the expression of tight junction proteins, which may ultimately lead to a chronic inflammatory state and chronic or progressive disease. Numerous studies have demonstrated that TJ complex damage and abnormal expression of TJ proteins (TJPs) are important causes of intestinal barrier dysfunction. For instance, ZO-1, occludin, and claudin have been found to be downregulated during the progression of IBD [44,45]. Metabolic syndrome-induced intestinal barrier attenuation can impair the compactness of the TJ complex in intestinal epithelial cells [46]. Regulation of epithelial barrier function by modulating the expression and localization of TJPs is consequently a potential new target for the treatment of these diseases.

### 2.3. Intestinal Microbiota

The bacterial amount is comparable to the number of total human cells, but they encode approximately 150 times more genes than the human genome [47]. Intestinal bacteria are referred to as the second genome of humans [48]. The intestinal flora refers to a complex composition and a large number of symbiotic microbial ecosystems that reside in the intestinal tract. The normally distributed intestinal microbiota has various physiological functions. The microbial content of the gastric was lower because pH, mucus concentration, and peristaltic action limited the growth of microorganisms. In the jejunum and ileum, microorganisms were dominated by *Firmicutes* and *Bacteroidetes*, respectively. However, in the colon, besides the ones mentioned, various microbiota such as *Actinobacteria*, *Proteobacteria*, and *Fusobacteria* co-exist [49]. The microbiota participates in energy-nutrient metabolism, removes toxic substances, prevents the invasion of harmful bacteria, and stimulates the establishment, improvement, and maintenance of immune function [50]. The biological barrier flora consists mainly of specialized anaerobic bacteria, such as *Lactobacillus* and *Bifidobacterium* [51]. They form a mycoderm by tightly binding to the intestinal mucosal epithelium through teichoic acid, which prevents pathogenic bacteria and their harmful substances from attacking [52]. The biological barrier can produce acetic acid, lactic acid, and short-chain fatty acids, maintaining an acidic environment for the inhibition of pathogenic bacteria habitation [53,54]. In recent years, fecal microbiota transplantation has been hotly researched, with the promise of repairing intestinal barrier damage through exogenous intake of probiotics and prebiotics [55,56].

It has been found that the intestinal flora also produces specific substances that enhance the immune function of the host mucosa. They promote the development and maturation of immune organs, improve specific and nonspecific immune functions, and have strong broad-spectrum antibacterial effects [57,58]. In addition to commensal flora, pathogenic and conditionally pathogenic flora are often present in the intestine. Certain densities of phages are also present in the mucus layer, and they lysed susceptible bacteria such as *Escherichia coli*, *Clostridium perfringens*, *Pseudomonas fragilis,* and *Enterococcus faecalis* [59,60,61]. Once the symbiotic flora is disturbed and the phage abundance decreases, the conditioned pathogenic bacteria will proliferate rapidly, causing harm to the organism. Intestinal flora disorders cause not only gastrointestinal diseases but also metabolic diseases, neuropsychiatric disorders, and cardiovascular diseases [62,63,64]. Therefore, an in-depth understanding of the interactions between the intestinal flora and host metabolism is important for the prevention and treatment of related diseases and for improving host health.

All of the components of the intestinal barrier are essential for the maintenance of intestinal health. Whether there is destruction of the intestinal epithelium, loosening of the tight junction proteins, or an imbalance in the intestinal flora, a variety of pathological conditions can result (Figure 1).

## 3. Pathological Mechanisms of Intestinal Barrier Disruption

Disruption of the intestinal barrier is often followed by specific pathological changes, such as oxidative/antioxidant imbalance, intestinal microecological disorders, and inflammatory factor/anti-inflammatory factor imbalance. These mechanisms subsequently lead to the development of multisystem diseases. Inflammation is a basic pathological process that occurs when intestinal tissues are stimulated by certain factors [65], which include biological factors such as bacteria and parasites, physical factors such as radioactivity and heat, and chemical factors such as drugs [66,67,68,69]. The overall consequence of the inflammatory response depends on the balance of pro- and anti-inflammatory factors. Proinflammatory cytokines (IL-1β, IL-6, and TNF-α) are released at an early stage and induce the subsequent cascade of inflammatory responses. Anti-inflammatory cytokines (IL-4, IL-10, and IL-13) can suppress inflammatory responses and prevent damage resulting from excessive inflammation. Immune cells and factors perform important roles in the onset and resolution of the inflammatory response. Localized inflammation is infiltrated by abundant immune cells (macrophages and neutrophils). The activation of immune cells and the release of inflammatory factors lead to an increase in the inflammatory response, while the reduction in the inflammatory response requires the release of anti-inflammatory factors by immune cells [70,71]. When inflammatory/anti-inflammatory factors are imbalanced, specific proinflammatory factors can bind to their receptors and promote activation of signaling targets such as NF-κB, JAK/STAT, and MAPK. They will then affect the expression of cytokines, adhesion molecules, immune receptors, and inflammation-related enzymes [72]. A further vicious cycle of inflammatory responses is created, which aggravates the disruption of the intestinal barrier. Blocking pathways associated with the imbalance in pro- and anti-inflammatory factors may be effective in suppressing intestinal inflammation.

Oxidative stress (OS) can be activated by abiotic and biotic factors such as radiation, drugs, and bacterial infections, leading to an inflammatory infiltrate of neutrophils [73,74,75]. OS refers to states where intracellular oxidation and antioxidation are unbalanced and tend toward overoxidation. Such an imbalance can cause inflammatory infiltration of neutrophils, increased secretion of proteases, and the production of large amounts of reactive oxygen species (ROS). OS can affect intestinal barrier function by participating in several physiological processes. These processes include slowing the regeneration of IECs, increasing the disruption of TJ, and reducing the secretion of antioxidants. The rapid renewal of IECs depends on the proliferation and differentiation of ISCs [76]. Certain concentrations of ROS in the intestine facilitate the differentiation of intestinal stem cells. However, a further increase in ROS concentration can lead to apoptosis [77]. Thus, OS would affect the renewal cycle of IECs. Hydrogen peroxide, a ROS, is commonly used in the establishment of cellular models of oxidative damage. In some experiments, it has been found to cause a reduction in the expression or translocation of TJ. This results in reduced transepithelial electrical resistance (TEER) and increased paracellular permeability [78,79]. This disruption subsequently leads to the progression of intestinal inflammation or even a systemic inflammatory response. OS can damage cells directly and activate various oxidative and antioxidative stress pathways that indirectly regulate the extent of tissue damage. Some recent studies of natural drugs have revealed that they exert anti-inflammatory and antioxidant effects by modulating oxidative stress-related signaling pathways such as the NF-κB, PI3K/Akt, and Nrf2 pathways. Most natural drugs work by increasing the expression of antioxidants, including superoxide dismutase (SOD), catalase (CAT), and glutathione (GSH), and reducing ROS levels [80,81,82]. They can even regulate the intestinal flora and promote the growth of probiotics such as *Lactobacilli* and *Bifidobacteria*, producing antioxidants to alleviate oxidative damage in the intestine [83,84,85]. OS, along with a diverse range of other pathophysiological processes, is involved in the regulation of intestinal barrier disruption (pre- and post-disruption).

The intestinal microbiota is an important and complex flora community. The microbiota mainly inhabits the external mucus layer and forms the microbial barrier of the intestine. Under physiological conditions, the body maintains intestinal microbial homeostasis through a variety of pathways. This homeostasis can be disrupted when the integrity of the intestinal barrier is threatened. When goblet cells and Paneth cells are massively damaged, mucus secretion decreases, and antimicrobial protein secretion is reduced. Consequently, the chances of intestinal microorganisms coming into direct contact with the enterocytes are increased. At the same time, the lysis of harmful bacteria becomes ineffective, and pathogenic bacteria are allowed to proliferate rapidly [86,87,88]. In addition, barrier damage may be accompanied by the destruction of M cells and dendritic cells. Antigen uptake, processing, and presentation cannot be performed, resulting in a failure to initiate the mucosal immune response [89,90]. The secretion of IgA against pathogenic bacteria by B cells in the intrinsic layer is reduced and fails to prevent the translocation of bacteria and their toxins [91,92]. Direct communication between the commensal intestinal flora and intestinal immunity controls disease progression [93]. Disruption of intestinal microbial homeostasis can disrupt the immune homeostasis of the intestinal mucosa, causing immune dysregulation and disease susceptibility. Abnormal alterations in both the amount and products of commensal flora lead to the activation of immune cells and cytokine overload, plus the activation of a range of receptors and proteins that ultimately contribute to the onset and progression of the disease [94]. Current research shows that dysbiosis of the intestinal flora is closely linked to the development of many diseases, such as IBD, irritable bowel syndrome, colorectal cancer, and antibiotic-associated diarrhea [95,96,97,98]. In this case, there is a change in the quality and quantity of beneficial, harmful, and conditionally pathogenic bacteria. Disruption of the intestinal barrier can lead to the displacement of intestinal microorganisms or their components into the body, producing local or systemic inflammation.

The onset of disease is often not limited to a single cause. In addition to the abovementioned pathological processes, intestinal ischemia/hypoxia, I/R, and individual extraintestinal pathological processes are also closely related to intestinal barrier damage. They do not have a clear causal relationship with intestinal barrier damage. Disruption of the intestinal barrier induces a variety of pathological processes, which in turn contribute to the disruption of the intestinal barrier. A complex interaction of mechanisms exists between them. Stress factors such as OS, bacteriophage cleavage of bacteria, and bacterial toxin secretion can cause intestinal flora disorders [99]. The inflammatory response can also exacerbate intestinal flora imbalances [100]. Following the disruption of the balance of intestinal microorganisms, the probiotics that are capable of modulating immune cells to exert immunosuppressive effects no longer dominate. Instead, the number and abundance of pathogenic bacteria increased. Along with the destruction of intestinal epithelial cells, the loosening of cell connections, and the increase in intestinal permeability, bacteria will translocate through the intestinal mucosa. Pathogen translocation proliferation further triggers an immune response through its specific structure and secreted metabolites [101]. Multiple pathological processes affect the destruction of the intestinal barrier. To end intestinal barrier damage, intervention at multiple links is necessary (Figure 2).

## 4. Signal Transduction Pathway

### 4.1. PI3K/Akt/mTOR Signaling Pathway

PI3K consists of the regulatory subunit p85 and the catalytic subunit p110 [102]. When binding to growth factor receptors, PI3K alters the protein structure of Akt and activates it [103]. Then, Akt activates or inhibits a series of downstream substrates, such as mTOR and TSC, by phosphorylation, thus regulating cell proliferation, differentiation, apoptosis, and other phenotypes. PI3K/Akt signaling is a canonical pathway in physiological and pathological processes [104,105]. This pathway is extensively involved in the proliferation, differentiation, and repair of intestinal epithelial cells. Zinc serves as an essential regulator of the intestinal barrier by activating the PI3K/Akt/mTOR signaling pathway. Its activation of this pathway not only upregulates TJPs’ expression and attenuates the hyperpermeable state of the intestinal barrier but also accelerates epithelial cell proliferation and promotes intestinal injury repair [106]. Currently, zinc carnosine is widely used in gastrointestinal mucosal injury, with good efficacy in chemoradiotherapy-induced esophagitis/enteritis, NSAID enteritis, Peptic ulcers, and other diseases [107,108,109], except for trace elements. As substrates of protein synthesis, amino acids also affect the growth, proliferation, and metabolism of epithelial cells. When the supplementation of amino acids is adequate, the activity of mTORC1 and the downstream production of proteins will be activated [110]. At the same time, the availability of amino acids affects the regulation of T-cell proliferation and activation, which consequently affects the clearance of infected cells by T-cells [111]. In contrast, these physiological processes are inhibited by amino acid deficiency [112]. Studies have long shown that glutamine supplementation can reverse the decrease in transepithelial resistance and increase permeability caused by deprivation. The mechanism of this role is the activation of the PI3K/Akt signaling pathways to regulate TJP expression and intercellular spatial function [113]. The mechanism may also be closely related to downstream mTOR signaling.

The PI3K signaling pathway is essential for maintaining the physiological homeostasis of the gut. Resveratrol is a polyphenolic stilbenoid present in several kinds of plants. Administration of resveratrol has been demonstrated to have many promising properties, such as resistance against inflammation, oxidative stress, cancer development, and neurodegeneration [114,115,116]. Zhuang et al. indicated that the mechanism by which resveratrol restores intestinal barrier dysfunction is closely related to the PI3K/Akt signaling pathway [81]. Resveratrol and its metabolite pretreatment can restore intestinal barrier dysfunction by inhibiting the inflammatory response, regulating intestinal microbial structure, alleviating OS-induced apoptosis of IECs, and increasing the expression of TJPs and secretion of mucus proteins in several ways [81,117,118]. There is another substance, ferulic acid, widely found in various plants, that has similar efficacy to resveratrol, but the mechanism of action is slightly different. Ferulic acid, a phenolic acid, can maintain the integrity of the intestinal barrier by inhibiting the expression of PTEN to increase the activity of the PI3K/Akt signaling pathway instead of activating PI3K/Akt and its downstream signaling directly [119]. In conclusion, both have the potential to be targeted drugs for treating intestinal diseases.

In previous studies, most drugs have been used to restore intestinal barrier dysfunction by modulating classical signaling pathways. In the classical PI3K/Akt/mTOR pathway, Akt acts as a nodal molecule that regulates various branches downstream or enables crosstalk with other pathways. However, we found that individual drugs exist to maintain intestinal function by modulating nonclassical pathways. In a recent study, it was demonstrated that the combination of asparagine and 3-indolepropionic acid synergistically activates Akt and subsequently downstream mTOR, thereby improving intestinal dysfunction and prolonging survival in mice suffering from traumatic colonic injury. However, unlike asparagine, the upstream regulatory molecule of Akt activation by 3-indolepropionic acid is not PI3K but PDK1 [120]. More interestingly, there are also individual substances that do not activate Akt but directly regulate mTOR through other molecules. Alpinetin, a flavonoid compound extracted from the seeds of Alpinia katsumadai Hayata, is able to reduce apoptosis of IECS through activation of the AhR/suv39h1/TSC2/mTOR signaling pathway [121].

Traditional Chinese medicine (TCM) has been proven to be effective via clinical trials for many types of diseases. In recent years, the signaling mechanisms underlying the effects of TCM have been gradually revealed. Huangqin decoction (HQD) has been employed to treat gastrointestinal diseases for thousands of years. Li et al. demonstrated that administration of HQD inhibits the Ras-PI3K-Akt-HIF-1α pathway and alleviates inflammatory responses in dextran sulfate sodium (DSS)-induced ulcerative colitis (UC) mice. The exhaustion of goblet cells and dysregulation of the intestinal microbiota can be significantly restored, further improving the status of the intestinal barrier [122]. Kuijieyuan decoction is another drug targeting UC based on the TCM knowledge system. It restores intestinal barrier injury in UC by inhibiting TLR4-dependent PI3K/Akt/NF-κB signaling. The levels of inflammatory and oxidative stress biomarkers are reduced. The relative abundances of beneficial and harmful bacteria are also rebalanced [123]. Akt, as a molecular switch of signaling pathways, interacts with multiple pathways to regulate intestinal barrier homeostasis. We found that the effect of some drugs is to restore intestinal function by activating PI3K/Akt pathways. However, there are also medications that restore intestinal health by inhibiting this pathway. This contradiction may be associated with the activation of different molecules downstream of Akt or with the interaction of intestinal diseases with other pathways at different stages of progression. The relationship between intestinal disease and this pathway needs to be further explored to lay the theoretical foundation for the development of TCM and targeted medicine.

### 4.2. MAPK Signaling Pathway

MAPK signaling universally exists in eukaryocytes and regulates proliferation, differentiation, and programmed cell death [124,125,126]. There are three main downstream pathways of MAPK, which include ERK, JNK, and p38/MAPK. These pathways participate in different physiological and pathological processes. Specifically, ERK is an important mediator of cell growth and differentiation [127]. MAPK/ERK signaling is predominantly modulated by Ras/Raf proteins [128]. ERK phosphorylation contributes to its activation and is transferred into the nucleus, affecting the expression of downstream genes. The biological functions of JNK and p38 are similar and are associated with inflammatory responses, cell growth, and apoptosis [129]. MAPK/JNK signaling can be induced by many types of environmental stress and inflammatory responses. Additionally, the stimulators of MAPK-JNK and MAPK-p38 partially overlap, such as TNF-α, IL-1, ultraviolet light, H_2_O_2,_ and heat shock [130,131,132].

Research on the relationship between MAPK signaling and intestinal barrier disorders has focused on IBD. Chlorogenic acid, an extract of honeysuckle, has antioxidative, anti-inflammatory, and antiviral properties [133]. A study has shown that chlorogenic acid improves the mitochondrial ultrastructure of intestinal epithelial cells. The relative levels of proinflammatory and anti-inflammatory factors are rebalanced [134]. This positive efficacy is closely associated with the regulation of MAPK signaling. In the 2,4,6-trinitrobenzene sulfonic acid (TNBS) and lipopolysaccharide (LPS)-induced colitis model used for IBD modeling, farrerol, a type of 2,3-dihydro-flavonoid obtained from rhododendron, can ameliorate the progression of colitis and reduce oxidative stress reactions through inhibiting multiple signaling pathways. The ERK and JNK pathways are important mediators [135]. N-Acetyl-seryl-aspartyl-lysyl-proline (AcSDKP) is a kind of tetrapeptide used for the suppression of inflammation and immune responses. It can inhibit the production of proinflammatory factors by decreasing the activity of MEK-ERK signaling [136].

JNK signaling has been shown to participate in the progression of apoptosis in intestinal epithelial cells induced by radiation. Bacterial-derived flagellating proteins can inhibit the JNK signaling pathway by upregulating the expression of mitogen-activated protein kinase phosphatase-7 (MKP-7) proteins and further restoring intestinal damage caused by radiation therapy [137]. This study provides strong evidence for the crosstalk between the host intestinal barrier and the microbiota. Impairment of the intestinal epithelial barrier and persistent intestinal inflammation are interlinked. The MAPK pathway is involved in the regulation of intestinal inflammation and the TJ state. Deoxynivalenol (DON) is a fungal toxin commonly present in cereal products. It can degrade ZO-1 and claudin-1, 3, and 4 and induce proinflammatory gene expression via MAPK. Along with the activation of the MAPK pathway, the expression of TJPs is decreased, mucosal permeability is increased, and intestinal inflammation is induced [129,138]. Resveratrol, which has JNK inhibitory activity, and berberine, which is used to treat intestinal infections, restore DON-induced intestinal dysfunction [139,140]. It has been confirmed that both improve intestinal function by inhibiting the NF-κB/MAPK signaling pathway. These studies suggest that the role of the MAPK signaling pathway in regulating intestinal homeostasis not only relies on immunomodulation and intestinal microecological balance but is also linked to the permeability of the intestinal tract. However, there is currently no approved drug to treat intestinal dysfunction caused by intestinal inflammation, immune response, or oxidative stress caused by various factors. There is also a need to further clarify the relationship between intestinal barrier repair and MAPK and to develop drugs with the ability to target therapeutics with precision.

### 4.3. AMPK Signaling Pathway

AMPK exists as a heterotrimeric complex consisting of an α-catalytic subunit, a β-regulatory subunit, and a γ-regulatory subunit. AMPK is expressed in various metabolism-related organs and is involved in physiological processes such as glucose metabolism, lipid metabolism, protein synthesis, and mitochondrial homeostasis [141,142,143]. Cellular glucose and long-chain fatty acid (LCFA) uptake are key physiological processes that regulate cellular energy homeostasis. AMPK is an essential nutrient sensor for maintaining energy homeostasis. Elevated levels of adenosine monophosphate, along with ATP and decreased levels of glucose, can activate AMPK [144]. On the one hand, activated AMPK can be translocated by phosphorylating glucose transporters. On the other hand, it can increase CD36 protein expression levels and membrane translocation in intestinal epithelial cells. This stimulates glucose uptake and catabolism while increasing the uptake of long-chain fatty acids [145,146]. Some studies have demonstrated that LCFAs are not only a source of energy but also a component of the cell membrane. They are also involved in the regulation of membrane protein function, which influences the immune response [147]. Recently, fatty acids combined with hepatocyte nuclear factor 4 were reported to be associated with the renewal of intestinal stem cells in mice [148]. When intestinal epithelial AMPK deficiency occurs, it may contribute to reduced epithelial regeneration and delayed repair after intestinal mucosal injury [149]. In addition, activation of AMPK enhances epithelial differentiation and TJ formation by promoting CDX2 expression, thereby maintaining intestinal barrier homeostasis by increasing transepithelial resistance and decreasing paracellular permeability [150,151]. These reports suggest that the AMPK signaling pathway not only participates in the regulation of energy metabolism, but also participates in the proliferation differentiation and renewal of intestinal epithelial cells.

Regulation of the AMPK pathway plays an instrumental role in the treatment of diseases related to intestinal barrier disruption, such as IBD, RE, and chemotherapeutic enterocolitis. 5-Aminosalicylic acid and sodium salicylate, commonly used in the treatment of IBD, alleviate the symptoms of DSS-induced colitis by activating the AMPKβ1 subunit of macrophages [152]. Radiation causes acute or chronic RE by destroying intestinal epithelial tissue, interfering with the reproduction of intestinal flora, and inducing metabolic disorders [153,154]. There is no uniform treatment strategy for RE. At present, physical defense and the application of radiation protective agents, intestinal probiotic preparations, and other drugs are still the main treatments [155]. Cordycepin, a natural nucleoside analog, activates AMPK to promote Keap1 degradation and Nrf2 dissociation, thereby preventing senescence, the aging of intestinal cells, and the occurrence of intestinal radioactive ulcers [156]. As the number of patients with abdominal and pelvic tumors increases, the occurrence of common radiotherapy complications of RE also rises. If the relationship between AMPK and RE can be clarified, a comprehensive treatment of physical defense plus targeted drug prophylaxis can be provided.

Intestinal I/R injury refers to a relatively aggressive pathological process that significantly impairs the maintenance of intestinal barrier integrity. Paeoniflorin has been reported to restore autophagy and ameliorate intestinal I/R injury by activating the LKB1/AMPK signaling pathway and promoting autophagosome synthesis [157]. Chemotherapy can also lead to intestinal inflammation due to intestinal ischemia. Chemotherapy induces the massive release of damage-associated model molecules (DAMPs), which induce excessive activation of the TLR-4/p38 pathway, followed by the release of large amounts of tissue factor (TF) [158,159]. Tissue factor refers to a kind of membrane-bound glycoprotein involved in exogenous coagulation pathways. Rapid upregulation of TF via exogenous coagulation pathways leads to intestinal thrombosis and ischemic injury, which ultimately leads to the development of chemotherapeutic enterocolitis [160]. In terms of irinotecan-induced chemotherapeutic enteritis, ozone inhibits TF expression by activating the AMPK/SOCS3 pathway, thereby alleviating chemotherapeutic enteritis caused by ischemic–hypoxic injury [161]. Clinical evidence exists for the treatment of chemotherapeutic oral mucositis with ozone [162]. Research on the mechanism of ozone relief for chemotherapy enteritis in mice provides new ideas and evidence for hyperbaric ozone autotransfusion therapy to treat intestinal damage caused by chemotherapy. AMPK is a known therapeutic target for glucose metabolism-related diseases, but research on its regulatory function in intestinal diseases is still limited and lacks sufficient depth. To apply the abovementioned drugs involved in AMPK regulation to clinical use, it is necessary to further deepen the research content and carry out large-scale clinical trials. This will provide new insights and a basis for new targeted treatments for intestinal mucosal destruction-related diseases.

### 4.4. JAK-STAT Signaling Pathway

The JAK-STAT signaling pathway was originally identified during interferon (IFN) transmission by Heim and Kerr et al. [163]. The JAK protein family contains four members, namely JAK1, JAK2, JAK3, and Tyk2. They can phosphorylate cytokine receptors and multiple signaling molecules containing specific Src homology 2 domains (SH2) [164]. Seven members of the STAT family have been found: STAT1-4, STAT5a, STAT5b, and STAT6. SH2 is a highly conserved and functional region among the multiple functional sites of STAT members [165]. It can bind heterologously to the receptor and indirectly affect the specificity of STATs in response to ligand signaling [163]. Classical JAK-STAT signaling begins with extracellular binding between the cytokine and its corresponding transmembrane receptor. The JAK-STAT signaling pathway is relatively straightforward in comparison with other signaling pathways. However, it is also widely involved in many important biological processes, such as cell proliferation, differentiation, apoptosis, and immune regulation.

In the acute intestinal inflammatory response, JAK/STAT1 signaling can activate R-ISCs and CBC ISCs during inflammatory injury. Activating these cells to enter the cell cycle accelerates ISC-mediated regeneration [166]. In addition, JAK/STAT signaling regulates the Notch signaling pathway in ISCs and facilitates mutual conversion between two different functions of ISCs, thus facilitating epithelial healing [167]. Peroxisome levels were found to be significantly higher in impaired crypts in patients with IBD, which represents a postinjury repair. By applying peroxisome proliferator to animal models, it was confirmed that this process was achieved by regulating JAK-STAT-SOX21A signals to promote stem cell differentiation [168]. However, the absence or abnormal activation of the JAK/STAT signaling pathway can also lead to oxidative stress, intestinal dysfunction, and excessive proliferation of ISCs. Fortunately, Flos Puerariae extract happens to inhibit the JAK-STAT pathway, restoring intestinal homeostasis and reducing ISC overproliferation [169]. In addition, it has various pharmacological effects, activating the Nrf2/keap1 signaling pathway to prevent cellular oxidative stress and reduce oxidative damage [169]. These studies are conducive to the development and utilization of related drugs that maintain the stable proliferation and differentiation of ISCs. This research also helps to provide more scientific treatment for patients with intestinal injuries with different pathological characteristics.

In the initial stages of intestinal mucosa attack by injurious factors, cell damage releases various inflammatory cytokines. Once the JAK-STAT signaling pathway is activated by cytokines, the body initiates innate immunity, coordinates the adaptive immune mechanism, and ultimately inhibits inflammation and the immune response [170]. However, a combination of injury factors often induces an over-mucosal immune response. Therefore, excessive penetration of the intrinsic layer by immune cells triggers the release of proinflammatory cytokines, causing different degrees of intestinal damage. Luteolin, Moringa oleifera seed extract, and Huangqi extract were found to have excellent anti-inflammatory effects in intestinal inflammation models. Mechanistically, they at least partially inhibit the activation of the JAK-STAT pathway, thereby reducing the production of key inflammatory mediators and inhibiting intestinal inflammation [171,172,173]. In addition, peptides extracted from Moringa oleifera seeds can also participate in the regulation of intestinal microorganisms and their metabolites to restore intestinal mucosal barrier integrity [172]. Along with intestinal inflammation, intestinal ischemia can cause serious intestinal damage. Inhibition of the JAK-STAT pathway can relieve intestinal mucosal cell apoptosis and intestinal dysfunction caused by ischemia [174,175]. Acupuncture plus moxibustion as a traditional TCM therapy has been proven to have a good effect on intestinal damage caused by autogenous orthotopic liver translation. Electroacupuncture pretreatment relieves the damage caused by intestinal ischemia, oxidative stress, and a severe inflammatory response after liver transplantation by inhibiting the JAK-STAT signaling pathway [176]. At present, individual oral small-molecule JAK inhibitors are available for the treatment of IBD patients or for clinical trials. However, few patients are suitable to receive these inhibitors, and they only have good efficacy in moderate to severe Crohn’s disease (CD). The drugs and treatments mentioned above can provide new ideas for the development of JAK inhibitors.

### 4.5. NF-κB Signaling Pathway

NF-κB refers to a family of transcription factors that contain p50 (NF-kappaB1), p52 (NF-kappaB2), p65 (RelA), RelB, and c-Rel [177]. These proteins can form homologous or heterologous dimers to regulate downstream gene expression and affect different biological processes. The major NF-κB protein complexes regulating the transcription of response genes are the p50/p65 heterodimer or the p50/p65 homodimer [178,179]. This complex is deactivated due to its combination with IκBα under normal conditions [180]. Bacterial and viral infection, inflammatory factors, and antigen–antibody combinations serve as inducers of NF-κB activation, as well as some other signaling pathways [181]. Despite distinct mechanisms, complexes of signaling proteins and ubiquitin polymers are subsequently formed in response to this activation process, further enhancing the activity of the inhibitor of the kappa B (IκB) kinase complex [182]. NF-κB dimers are released from the complex after the phosphorylation of IκB. This type of NF-κB with robust protein activity is transmitted to the cell nucleus, binds to specific sequences at DNA sites, and initiates the transcription of genes [183].

NF-κB is an important mediator of the inflammatory response. As a transcription factor at the early stage, NF-κB activity changes correspondingly in response to damage to intestinal epithelial cells. The abrupt release of cytokines induced by intestinal barrier destruction can facilitate the transcription-promoting activity of NF-κB and the synthesis of proinflammatory factors. This signaling abnormality has been proven to be an important contributor to IBD and RE [184,185]. *Canna x generalis* L.H. Bailey rhizome extract is a kind of natural compound. In DSS-induced UC mice, its administration can attenuate barrier dysfunction, oxidative stress, and the inflammatory response by targeting the TLR4/NF-κB and NLRP3 inflammasome pathways [186]. Brucea javanica oil emulsion is a traditional Chinese folk medicine commonly used for dysentery and IBD. Huang et al. employed TNBS to generate a rat model with CD. Treatment of this model with Brucea javanica oil emulsion can effectively ameliorate systemic and colonic signs of CD by inhibiting the TLR4/NF-κB signaling pathway [187]. Salazosulfapyridine and azathioprine are both mainstay drugs for the alleviation of IBD progression. The authors also showed that the efficacy of Brucea javanica oil emulsion outperformed these two classical drugs [188]. Moreover, as mentioned above, regulation of NF-κB is an important mechanism by which Huangqin and Kuijieyuan decoctions mitigate inflammation and improve intestinal barrier function, in addition to their effects on Akt signaling [122,123].

Local radiation can significantly disturb microbiota symbiosis, inducing the release of endotoxin and alterations of the signaling cascade [189,190]. The effects of treatment with a single kind of antibiotic may be unsatisfactory. However, the administration of an antibiotic cocktail has exhibited great progress in the prevention and therapy of intestinal damage induced by irradiation. This regimen can not only reduce the relative abundances of pathogenic bacteria and restore the composition of the intestinal bacteria but can also alleviate intestinal inflammation in an NF-κB-dependent manner [191]. In addition to exogenous compounds, specific types of gastrointestinal hormones play important roles in maintaining the intestinal barrier. Ghrelin, known as the ‘hunger hormone,’ plays a role in growth hormone release, food intake, and fat deposition [192]. Existing studies have demonstrated that exogenous supplementation or endogenous upregulation of ghrelin can restore the functions of the intestinal barrier and promote the regeneration of the intestinal epithelium and ISCs. The deterioration of the damaged barrier caused by irradiation is reversed by activating Notch signaling [193]. It is thus suggested that a combination of ghrelin and drugs with NF-κB inhibition effects can be used to treat radioenteritis. The intestinal epithelium is maintained while inhibiting the inflammatory response. Activation of the NF-κB pathway is involved in the pathogenesis of a variety of chronic inflammatory diseases. Many drugs, natural products, and normal or recombinant proteins that inhibit NF-κB activation are expected to be used in the treatment of NF-κB-related diseases.

In addition to the signal transduction pathways and various drugs described above, there are many traditional Chinese medicines that restore intestinal barrier function by modulating different pathways (Table 1). The signaling molecules in each of these pathways have potential as targets for the treatment of diseases associated with intestinal barrier disruption.

## 5. Methods of Literature Review

A PubMed search using the search terms “intestinal barrier” and “signal transduction pathway”, which apply AND techniques and are limited to the last decade of publication, yielded 707 results. Other PubMed searches of “((intestinal barrier) AND (signaling transduction pathways) AND (y_10 [81])) AND (therapy)” returned 311 results. The addition of the keyword “pathophysiology” to the previous list yielded 21 results. In addition, articles from the past decade were searched by the keywords “intestinal barrier”, “signal transduction pathway”, and “traditional Chinese medicine”, ultimately obtaining 71 results. Relevant resources from selected articles were also utilized as needed.

Such a search would include pathophysiological variations and novel therapeutic options for intestinal barrier dysfunction associated with signaling transduction pathways. Interestingly, after sorting and organizing the results, we found that among the various signaling pathways related to the intestinal barrier, the aforementioned five signaling pathways were the most widely studied. Moreover, numerous drugs with the potential to target the above five signaling pathways either individually or in interaction with each other to repair or maintain the integrity of the intestinal barrier are available. About 222 articles were eventually cited in this review. We focus on the therapeutic mechanisms of potentially targeted drugs capable of restoring intestinal health. Hopefully, novel targeted drugs or novel therapeutic options will be explored for the treatment of various diseases associated with intestinal barrier disruption.

## 6. Limitations and Challenges

### 6.1. The Nature of Animal Model

TNBS- or DSS-induced animal models are most commonly used for research on intestinal inflammation, especially IBD. Despite the partial overlap of the model characteristics with human diseases, there are some drawbacks to these classical models. First, there are differences between humans and mice. Humans and mice both fall under the class of mammals and have great similarities in genetic and physiological features. However, numerous studies have revealed the distinctions between mouse and human intestines. It is reasonable to hypothesize that there are differences in the pathological mechanisms concerning intestinal barrier damage. Moreover, the preparation processes for the chemical induction of intestinal inflammation require only several days or weeks, which cannot thoroughly simulate the pathological progression of IBD. Some studies have investigated the signaling alterations and drug efficacy of intestinal inflammation using intestinal epithelial cells or immortalized colon cancer cells. However, the in vivo intestinal system is absent, further impairing the reliability of the article’s data. In summary, there are apparent flaws in the experimental models in this research field. More concerns about translational application and repeatability in humans should be raised.

### 6.2. The Physiological Differences between the Intestines

The intestine contains the small intestine, colon, and rectum. In addition to the morphological and functional differences among these parts, microbiota symbiosis, epigenetics, and auxanology have also been proven to have significant differences [210,211]. The different sections of the intestine may undergo pathological progression with some distinctions if they are subjected to specific injuries and signaling pathways. However, most articles have neglected this feature. Some studies did not accurately define the intestinal locations or discuss the natural differences. Even the specimens collected from animal models were not clearly defined. These drawbacks can generate potential experimental bias and affect the reliability of conclusions. Further research should pay more attention to this unique characteristic, especially for in vivo experiments.

### 6.3. Lack of Effective Drugs

Drugs for therapy against intestinal barrier damage can be summarized as mucosal protective agents, antibiotics, probiotics, immunosuppressive agents, and various biological agents. Although these drugs have shown definite efficacy in alleviating disease development, some shortcomings have been exposed during clinical practice. For instance, 5-ASA is the first-line agent for inducing remission in mild or moderate IBD patients. Conventional oral 5-ASA is absorbed in the upper part of the small intestine, while the drug concentration that reaches the colon is low. Therefore, higher doses are usually required to ensure drug efficacy. These higher doses can cause adverse effects such as fever, diarrhea, and nephrotoxicity [212,213]. Systemic corticosteroids can be used for mildly active patients who have poor responses to 5-ASA therapy or moderate to severe degrees of disease. Nevertheless, long-term hormone use can cause osteoporosis and femoral head necrosis, which can also induce or aggravate infections and delay tissue healing. Immunosuppressive therapy is accompanied by decreased blood counts and an increased risk of lymphoma. Several JAK inhibitors have been approved for autoimmune diseases, but only tofacitinib and upadacitinib have been approved for IBD. The interest in tofacitinib for the treatment of UC has slowed due to safety concerns such as increased herpes zoster infection and elevated blood lipid levels [214]. In clinical trials, upadacitinib, which is a JAK1 inhibitor with high specificity, also failed to reduce the risk of infection and thrombosis and increased creatine phosphokinase [215,216]. With the study of the mechanisms of various intestinal mucosal disruption diseases, more biologics and small-molecule drugs targeting immune targets are also needed. Research on these targeted drugs with translational potential may provide a remedy for the repair of intestinal mucosal damage with drugs that have stable efficacy and simultaneously have low toxic side effects.

## 7. Future Perspectives

The efficacy of first-line agents in IBD and RE is still limited. In recent years, considering the unsatisfactory performance, high cost-effectiveness, and significant side effects of artificially targeted drugs, drug companies have emphasized the development of natural compounds and herbal medicines. This review introduces the existing achievements concerning their application in treating intestinal barrier disorders. They can function as effective drugs by targeting various kinds of signaling pathways. The advantages of natural compounds and herbal medicines over synthetic targeted drugs are as follows: (i) Accessibility. Natural compounds and herbal medicines are extracted from plants and other natural sources, which means that the preparation processes and acquisition of raw materials are relatively easy, contributing to controlling the production cost and technical threshold. (ii) Verification. The compositions of herbal medicines are orchestrated based on TCM theory. Their efficacy has been verified by numerous recipients. The use of a single kind of natural compound has also been established in these ancient prescriptions. The long history can be analogous to large-scale real-world clinical trials. Their wide application suggests their effectiveness and safety. Therefore, the research value and probabilities of clinical translation are much higher than those of novel synthetic drugs. (iii) Acceptance. Patient compliance is a critical factor in determining actual efficacy, which largely depends on the drug’s acceptance by recipients. If patients recognize effectiveness and safety, they naturally tend to insist on drug administration. Natural compounds and herbal medicines precisely meet patients’ expectations due to long-term verification and natural sources. Artemisinin used for malaria therapy serves as assertive evidence of this. Researchers should focus on the clinical translation of natural compounds and herbal medicines, which may become a promising direction for intestinal disorder treatment.

Most current studies have focused on the efficacy verification and corresponding mechanisms of a single drug. In consideration of the tendency toward combinational drug administration and complex signaling crosstalk in response to intestinal barrier disorders, the development of efficient combination medications may be a promising direction, which originates from two trains of thought. One is a signaling mechanism. Accumulating studies have revealed the mechanistic interactions between different signaling pathways in the intestines [217,218,219]. For instance, coregulation of PI3K/Akt and IκBα/NF-κB signaling contributes to the restoration of oxidative stress-induced intestinal epithelium. This evidence has shown the importance of multiple signaling interferences. Inactivators or inhibitors targeting interacting pathways may enhance the efficacy of a single drug. Future research can be conducted on the basis of previous reports about signaling mechanisms. On the other hand, the mainstay regimen against intestinal barrier disorders cannot meet clinical requirements. However, this does not thoroughly deny their significance and means that these drugs should be replaced. As introduced above, TCM exhibits great potential in the treatment of intestinal barrier dysfunctions with excellent accessibility and safety. Combinational application of some natural compounds and traditional Chinese decoctions serves as another approach to augmenting efficacy. Taken together, research on the exploration of combination administration is urgently required based on the existing knowledge concerning the mechanisms underlying intestinal disorder progression.

Clinical trials are requisite for the entry of novel drugs into medical practice. Infliximab, a TNF inhibitor, has been widely used for the treatment of autoimmune diseases, including IBD. Despite its definite functions in IBD treatment, a great number of recent clinical trials have been conducted to test the efficacy and identify the best indications for infliximab administration [220,221,222]. To promote the translational application of new regimens, experiments on mice and cell models are insufficient. Only well-designed, large-scale, multicenter clinical trials can detect the actual effects of these drugs or regimens. Considering the specific location of the intestines, oral administration may serve as the fastest and most direct route among all modes of drug delivery. Effective oral drugs preferentially entering clinical trials are encouraged. The population heterogeneity needs further attention, which has been predominantly shown in the application of cancer-targeted drugs. Caution should be taken when generalizing drug application across areas and subpopulations, and clinical trials based on specific populations should be conducted. Furthermore, long-term administration safety should be carefully evaluated to guarantee the value of clinical translation.

## 8. Conclusions

In this review, we discuss the relationship between multiple signaling pathways and the regulation of the pathophysiological state of the intestinal barrier. Failure of intestinal barriers can be caused by various factors, which can lead to a wide range of diseases and complex conditions. Existing therapeutic methods have limited efficacy and are accompanied by multiple complications. Recent studies have focused on the effects of natural extracts and metabolites on the restoration of intestinal barrier function. This is mainly due to their easy availability, low cost, and high safety. Some of these drugs are capable of simultaneously regulating multiple dysregulated signaling pathways to restore gut barrier function. Alternatively, several drugs can be combined to restore mechanical, chemical, microbial, and immune functions through different signaling pathways. The contribution of these in vivo and in vitro experiments to the identification of disease causes and pathogenesis and the formulation of treatment programs is undisputed. However, there are still some limitations to the various studies. Nevertheless, with this as the theoretical foundation and scientific basis, further improvement in drug combination protocols and large-scale clinical trials are expected to provide multiple safe and effective novel targeted therapeutic options for intestinal barrier disruption diseases. A comprehensive understanding of the intestinal barrier and signal transduction pathways will help to achieve better intestinal health and improve quality of life.

## Figures and Tables

**Figure 1 pharmaceuticals-16-01216-f001:**
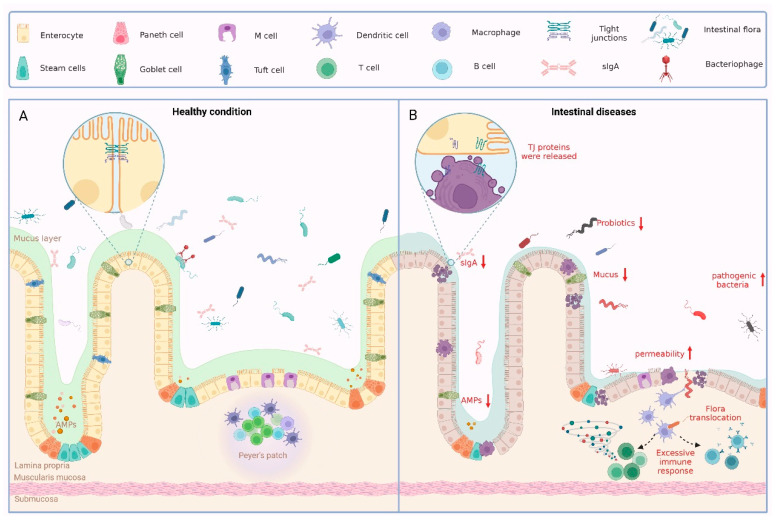
Physiological and pathological status of the intestinal barrier. (**A**) Large numbers of intestinal flora and some phages in the intestinal lumen together form a biological barrier. Various IECs form a solid mechanical barrier through intercellular junctions. IECs secrete mucus to form a chemical barrier on the inner wall of the intestinal cavity, which has the effect of preventing bacteria from remaining on the intestinal wall. The immune barrier is formed by the lymph nodes of the intestinal wall and the lymphocytes in the lamina propria of the mucosa. Together, these four elements form the complete intestinal barrier. (**B**) When the intestinal barrier is disrupted for several reasons, pathological states such as disturbances in the intestinal flora, reduced secretion of intestinal epithelial cells, loosening of TJ, and excessive immune responses occur.

**Figure 2 pharmaceuticals-16-01216-f002:**
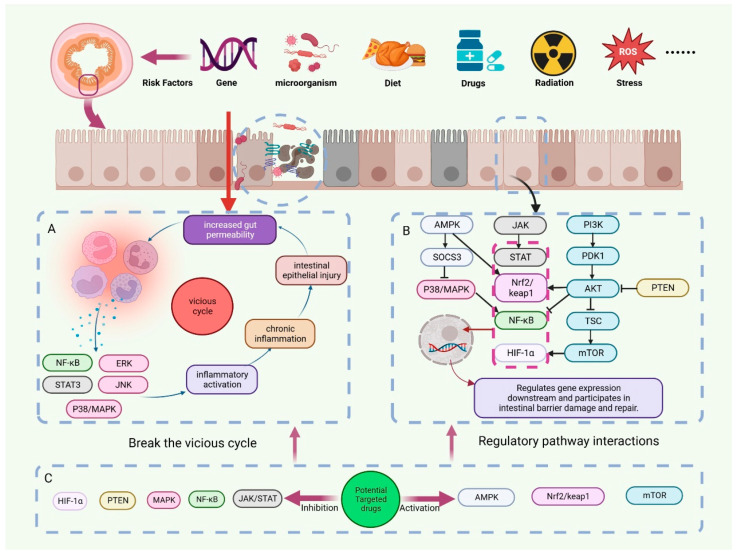
Regulation of intestinal barrier disruption by potential targeting drugs. Genetic susceptibility factors, radiotherapy, drugs, oxidative stress, and other risk factors can cause disruption of the intestinal barrier. (**A**) A vicious cycle of immune response-IEC damage occurs after intestinal barrier disruption. (**B**) Multiple signal transduction pathways interact extensively in the disruption and repair of the intestinal barrier. (**C**) We have summarized numerous drugs that have the potential to target the treatment of intestinal barrier disruption. They can regulate signal transduction pathways by inhibiting targets such as PTEN and MAPK or activating targets such as AMPK and mTOR, thereby breaking the vicious cycle and repairing the intestinal barrier.

**Table 1 pharmaceuticals-16-01216-t001:** Modulation of the intestinal barrier by diverse traditional Chinese medicines.

Medicine	Signaling Pathway	Activation or Inhibition	Intestinal Pathologies	Effects	References
Compound sophorae decoction	Notch	Inhibition	UC	Reduced the loss of muc-2 and goblet cells. Repairs damage of the epithelial cell barrier and prevents mucosal barrier damage, thus improving permeability.	[194]
QingBai decoction	NF-κB and Notch	Inhibition	UC, CD	Inhibits the expression of pro-inflammatory cytokines and prevents the loss of muc-2	[195]
Shaoyao decoction	MKP1/NF-κB/NLRP3	Inhibition	UC	Regulates inflammatory factor homeostasis. Inhibits inflammation, necrosis, and pyroptosis.	[196]
Gardenia decoction	NF-κB	Inhibition	Endotoxin-induced intestinal mucosal injury	Inhibits pro-inflammatory cytokines and up-regulates total antioxidant capacity.	[197]
Baitouweng decoction	IL-6/STAT3	Inhibition	UC	Inhibits the release of classical inflammatory factors such as interleukin 6 and STAT3 and regulates the structure of the intestinal flora.	[198]
Gegen Qinlian decoction	IL-6/JAK2/STAT3	Inhibition	UC	Regulates the homeostasis of Th17/Treg cells in colon tissue and inhibits the progression of the inflammatory response.	[199]
Huangqin decoction	PI3K/AKT	Inhibition	UC	The intact mucosal layer structure of the intestinal epithelial mucosa was maintained, reducing the production of pro-inflammatory cytokines and improving the infiltration of inflammatory cells.	[200]
Ligularia fischeri root extracts	Bcl-2/Bax	Activation	Diarrhea disease	Inhibit apoptosis of intestinal epithelial cells and release of pro-inflammatory cytokines.	[83]
Huang Bai Jian Pi decoction	PI3K/AKT/NF-κB	Inhibition	UC	Regulates the balance of inflammatory and anti-inflammatory factors, significantly improving diarrhoeal symptoms and relieving the systemic inflammatory response.	[201]
Huanglian Jiedu Decoction	JAK2/STAT3, NF-κB and Nrf2	Inhibition (JAK2/STAT3, NF-κB) Activation (Nrf2)	Acute UC	Reducing the secretion of inflammatory factors and peroxidases, promoting the proliferation of intestinal epithelial cells, and reducing the risk of intestinal fibrosis.	[202,203]
Gegen Qinlian decoction	Notch	Activation/Inhibition	Acute/chronic UC	Promotes the proliferation, differentiation, and secretion of epithelial cells to regulate the homeostasis of the colonic mucosa through bidirectional regulation.	[204]
Huangqin decoction	IFN-γ/JAK/ETS	Inhibition	UC	Inhibits the release of inflammatory factors and apoptosis of intestinal epithelial cells.	[205]
Houttuynia cordata Thunb decoction	MAPK (ERK1/2)	Inhibition	Colitis	Alleviation of intestinal inflammation by increasing the expression of tight junction proteins and anti-inflammatory factors.	[206]
Zhizhu Decoction	SIRT1/FoxO1	Activation	Slow transit constipation	Increased expression of TJ proteins and MUC2, significantly relieving intestinal inflammation and oxidative stress levels.	[207]
Qushi Huayu decoction	MAPK	Inhibition	Intestinal leakage	Regulates the composition of the gut microbiota and promotes the expression of TJ proteins.	[208]
Sanhuang Xiexin decoction	TLR4-MyD88-NF-κB	Inhibition	UC	Inhibit inflammatory cytokines, reduce oxidative stress levels, and restore intestinal probiotic abundance.	[209]

## Data Availability

Data sharing is not applicable.

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
