# Peer review of "Critical Signaling Transduction Pathways and Intestinal Barrier: Implications for Pathophysiology and Therapeutics"

_pharmaceuticals, 2023, doi:10.3390/ph16091216_

Round 1
Reviewer 1 Report
The manuscript entitled “Signaling transduction pathway and intestinal barrier: implications for pathophysiology and therapeutics” gives a comprehensive summarize on the relationships between the intestinal barrier and signaling pathways. This paper also discusses the limitations and prospects of targeted therapies with translational potential, providing new horizons for restoring intestinal barrier function. After reading this paper, I think this paper is complete and contains all the necessary components. The structure is reasonable and moderately illustrated. However, there are some problems to be further improved as well:
1. Pay particular attention to English grammar and sentence structure (lines 44-45, line 111, lines 136-137, et al.) so that the meaning of the paper is clear to the reader. I would suggest the authors get editing help from someone with full professional proficiency in English.
2. Part 2.2: Please keep the abbreviations consistent. Tight junctions, TJs or TJ?
3. Line 111: “TJs contain a variety of proteins…”, contain? Not appropriate.
4. Lines 132-133: Please provide reference(s) to backup that statement (also lines 275-276).
5. Lines 352-355: It is incomplete to cite references (also lines 374-376). Please check throughout the full text.
6. A review is not a list of the work of others, and it requires unified thinking. Also, there should be a rigorous logical relationship between sentences. Please revise the full text to reflect the logical connection of the sentence or paragraph (e.g. using connectives).
7. Lines 380-382: The description of basic theory and research progress must ensure stringency. “…no drug”? Rephrase, for example “there is currently no available drug approved to treat…”.
8. Line 458: Please change “However” to “However,”.
9. The discussion is not deep enough. Please try to give more detailed mechanistic insight. For example, part 3 (line 172) can add some content about immune cells changes when the intestinal barrier is broken.
10. I suggest you present the conclusions or ideas of other studies in a more cohesive and concise way (Master variable + mechanism + phenotype + disease). Also, how do the findings relate to targeted therapies for restoring intestinal barrier function. My suggestion is that you can rewrite the findings of other studies in the citations section of the text.
Extensive editing of English language required
Author Response
Responses to the reviewer1 comments
1.Pay particular attention to English grammar and sentence structure (lines 44-45, line 111, lines 136-137, et al.) so that the meaning of the paper is clear to the reader. I would suggest the authors get editing help from someone with full professional proficiency in English.
Response: Thanks for your great advice for the English grammar and sentence structure. We have revised the manuscript carefully according to the “Grammar Instructions”. And, the manuscript has been checked and corrected by a professional editing company.
2.Part 2.2: Please keep the abbreviations consistent. Tight junctions, TJs or TJ?
Response: Thanks for your valuable comments. Tight junction are abbreviated as TJ and Tight junction proteins as TJPs. Relevant modifications have been reflected in the manuscript
3.Line 111: “TJs contain a variety of proteins…”, contain? Not appropriate.
Response: We feel sorry about the unappropriated word. We have changed“TJs contain a variety of proteins…”to“TJs include a variety of proteins…” in our manuscript.
4.Lines 132-133: Please provide reference(s) to backup that statement (also lines 275-276).
Response: Thanks for your valuable suggestion. We have provide reference to backup the statement for lines 132-133. DOI: 10.3748/wjg.v29.i1.19. For lines 275-276, this sentence is the general statement, explained by the next sentence and supported by references
5.Lines 352-355: It is incomplete to cite references (also lines 374-376). Please check throughout the full text.
Response: We feel sorry for our careless work. We are not sure if we fully understood what you meant, and we have checked and corrected all the references in the manuscript according our understanding. If it does not meet your requirements, we kindly ask you to contact us again to revise it.
6.A review is not a list of the work of others, and it requires unified thinking. Also, there should be a rigorous logical relationship between sentences. Please revise the full text to reflect the logical connection of the sentence or paragraph (e.g. using connectives).
Response: We apologize for the fact that our manuscript does suffer from a lack of logical clarity. We have revised the manuscript in several places by adding conjunctions and rewriting to make it more logical. Thank you for your valuable comments.
7.Lines 380-382: The description of basic theory and research progress must ensure stringency. “…no drug”? Rephrase, for example “there is currently no available drug approved to treat…”.
Response: Thanks for your valuable suggestion. We feel sorry about the lack of stringency in the language. We have revised the statements in the manuscript based on your comments.
8.Line 458: Please change “However” to “However,”.
Response: Thanks for your careful check for our draft. We have changed “However” to “However,” in our manuscript.
9.The discussion is not deep enough. Please try to give more detailed mechanistic insight. For example, part 3 (line 172) can add some content about immune cells changes when the intestinal barrier is broken.
Response: We are sorry for the problems that are real in the manuscript. Immune defense being an important part of maintaining the integrity of the intestinal barrier, there is no doubt that immune cells are changed after the disruption of the intestinal barrier. We have revised and added to this section of the manuscript. Thank you very much for your valuable comments.
10.I suggest you present the conclusions or ideas of other studies in a more cohesive and concise way (Master variable + mechanism + phenotype + disease). Also, how do the findings relate to targeted therapies for restoring intestinal barrier function. My suggestion is that you can rewrite the findings of other studies in the citations section of the text.
Response: In this manuscript, we have cited many meaningful research findings. However, we feel very sorry that we did not present the research results of others in a more coherent and concise manner during the citation process. We have revised and added to the full text in accordance with your comments to make the relationship of these results to targeted therapies for restoring intestinal bowel barrier function clearer. Thanks again for your valuable comments.
Reviewer 2 Report
The manuscript submitted by Gao et al., titled: "Signaling transduction pathway and intestinal barrier: implications for pathophysiology and therapeutics" is an interesting review discussing the relationship between certain signal transduction pathways and the intestinal barrier relationship. The review is interesting and the reviewer would like to offer some points for the authors to consider below:
1. The title of the manuscript would benefit from modification so as to be more specific and accurate. Not all signaling pathways are investigated/discussed in the manuscript so this should be reflected in the title.
2. It would be important to consider a special paragraph on the rationale for the pathways selected and possible graph or other pictogram or table with the organized rationale for the review.
3. A short paragraph discussing selection and identification criteria used for the selected literature considered in the review would strengthen the paper.
4. The paper would benefit from a discussion regarding the microbiome and signaling as well as certain know nutrients typically found in the diet and with established signaling properties (namely amino acids). Here are a couple of manuscripts which may be found useful in this regard:
- Sikalidis AK (2015) Amino Acids and Immune Response: A role for cysteine, glutamine, phenylalanine, tryptophan and arginine in T-cell function and cancer? Pathol Oncol Res. 21(1):9-17. doi: 10.1007/s12253-014-9860-0.
-
- Sikalidis AK, Maykish A (2020) The Gut Microbiome and Type 2 Diabetes Mellitus; discussing a complex relationship. Biomedicines. 8(1):8. doi.10.3390/biomedicines8010008.
Nice work overall.
English language is OK overall the manuscript would benefit from proofreading by a native English speaker.
Author Response
Responses to the reviewer2 comments
1.The title of the manuscript would benefit from modification so as to be more specific and accurate. Not all signaling pathways are investigated/discussed in the manuscript so this should be reflected in the title.
Response: We are sorry for the lack of rigorous selection of the title, which we have changed to “Critcal signaling transduction pathways and intestinal barrier: implications for pathophysiology and therapeutics”. Thanks for your valuable comments.
2.It would be important to consider a special paragraph on the rationale for the pathways selected and possible graph or other pictogram or table with the organized rationale for the review.
Response: The selection of signaling transduction pathways is indeed an important part of this manuscript, and I apologize for our carelessness. We have added a "Methods of Literature Review" section to the manuscript to make this review more logical. Thanks for your valuable comments.
3.A short paragraph discussing selection and identification criteria used for the selected literature considered in the review would strengthen the paper.
Response: We are also aware of this problem, and we have provided a detailed description of citation filtering and citation identification in the "Methods of Literature Review" section. Thank you again for your valuable comments.
4.The paper would benefit from a discussion regarding the microbiome and signaling as well as certain know nutrients typically found in the diet and with established signaling properties (namely amino acids). Here are a couple of manuscripts which may be found useful in this regard:
- Sikalidis AK (2015) Amino Acids and Immune Response: A role for cysteine, glutamine, phenylalanine, tryptophan and arginine in T-cell function and cancer? Pathol Oncol Res. 21(1):9-17. doi: 10.1007/s12253-014-9860-0.
- Sikalidis AK, Maykish A (2020) The Gut Microbiome and Type 2 Diabetes Mellitus; discussing a complex relationship. Biomedicines. 8(1):8. doi.10.3390/biomedicines8010008.
Response: Thank you for your valuable suggestions. We apologize that the literature search was not comprehensive enough and we have cited the two excellent articles mentioned above on page 8 and page 1 of the manuscript, respectively.
Reviewer 3 Report
The authors have composed a well-written, thorough review of a topic of current interest to many scientists and health professionals. The paper is organized and detailed.
My suggestions are for describing some topics in more detail and including several additional references.
Line 100-: Another reference you could include is: Howitt et al., Tuft cells, taste-chemosensory cells, orchestrate parasite type 2 immunity in the gut.Science351,1329-1333(2016).DOI:10.1126/science.aaf1648
Line 124: "ectopic expression of TJ proteins are important causes of intestinal barrier dysfunction." This is interesting. Can you provide an example with citation?
Line 136: "Normal distributions of intestinal microbiota have various physiological functions." Perhaps here it would be important to state that there are different microbiota profiles along the different sections of the GI tract.
Line 158: "Any of the components..." Better to state, "All of the components..."
Line 192: Oxidative stress (OS) can be activated by biotin, radiation, drugs and bacterial infection, resulting in inflammatory infiltration of neutrophils[63-65]." Which of these citations addresses biotin?
Line 352, 363: "proven": Line 367 "proof"; and elsewhere: Instead, use the terms "supports," "shown" or "provides evidence."
In an appropriate section, perhaps under Future Perspectives, include mention of fecal transplantation for repairing barrier damage. For example: Cheng S, Ma X, Geng S, Jiang X, Li Y, Hu L, Li J, Wang Y, Han X. Fecal Microbiota Transplantation Beneficially Regulates Intestinal Mucosal Autophagy and Alleviates Gut Barrier Injury. mSystems. 2018 Oct 9;3(5):e00137-18. doi: 10.1128/mSystems.00137-18. PMID: 30320222; PMCID: PMC6178585.
The English language is generally very good.
Author Response
Responses to the reviewer3 comments
1.Line 100-: Another reference you could include is: Howitt et al., Tuft cells, taste-chemosensory cells, orchestrate parasite type 2 immunity in the gut.Science351,1329-1333(2016).DOI:10.1126/science.aaf1648
Response: Thanks for your valuable suggestions. we have attached this reference in the corresponding paragraph.
2.Line 124: "ectopic expression of TJ proteins are important causes of intestinal barrier dysfunction." This is interesting. Can you provide an example with citation?
Response: Increasing studies have demonstrated that TJ complex damage and abnormal expression of TJ proteins are important causes of intestinal barrier dysfunction. For instance, ZO-1, occludin and claudin have been found to be downregulated during the progression of IBD. Metabolic syndrome-induced intestinal barrier attenuation can impair the compactness of the TJ complex in intestinal epithelial cells. In the manuscript, we utilize the following three articles to support these points.
1.Qiu, S., et al., Maresin 1 alleviates dextran sulfate sodium-induced ulcerative colitis by regulating NRF2 and TLR4/NF-kB signaling pathway. Int Immunopharmacol, 2020. 78: p. 106018.
2.Li, B.L., et al., Luteolin alleviates ulcerative colitis through SHP-1/STAT3 pathway. Inflamm Res, 2021. 70(6): p. 705-717.
3.Yang, S., et al., Akebia saponin D ameliorates metabolic syndrome (MetS) via remodeling gut microbiota and attenuating intestinal barrier injury. Biomed Pharmacother, 2021. 138: p. 111441.
I apologize, and I think perhaps our wording may have caused you to misinterpret this sentence. We intended to state "aberrant expression of tight junction proteins" not expression of tight junction proteins in other tissues or organs. We have changed the manuscript to "aberrant expression of tight junction proteins". Thank you for your valuable comments.
3.Line 136: "Normal distributions of intestinal microbiota have various physiological functions." Perhaps here it would be important to state that there are different microbiota profiles along the different sections of the GI tract.
Response: Thanks a lot for your valuable comments. There are indeed differences in the distribution of microbiota in different parts of the GI tract, and we have added this important section in the manuscript under "2.3. Intestinal microbiota".
4.Line 158: "Any of the components..." Better to state, "All of the components..."
Response: Thanks for your careful reading and valuable comment. We have revised "Any of the components..." to "Any of the components..." in our manuscript.
5.Line 192: Oxidative stress (OS) can be activated by biotin, radiation, drugs and bacterial infection, resulting in inflammatory infiltration of neutrophils[63-65]." Which of these citations addresses biotin?
Response: I apologize for the clerical errors in our manuscript. We wanted to express "biotic" and "abiotic", but carelessly wrote biotic as "biotin", which led you to interpret the sentence in a different manner. Thank you very much for your careful review of the manuscript, which we have changed to read " Oxidative stress (OS) can be activated by abiotic and biotic factors such as radiation, drugs, and bacterial infections leading to an inflammatory infiltrate of neutrophils".
6.Line 352, 363: "proven": Line 367 "proof"; and elsewhere: Instead, use the terms "supports," "shown" or "provides evidence."
Response: Thanks for your careful reading and valuable comment. We have revised the manuscript based on your suggestions.
7.In an appropriate section, perhaps under Future Perspectives, include mention of fecal transplantation for repairing barrier damage. For example: Cheng S, Ma X, Geng S, Jiang X, Li Y, Hu L, Li J, Wang Y, Han X. Fecal Microbiota Transplantation Beneficially Regulates Intestinal Mucosal Autophagy and Alleviates Gut Barrier Injury. mSystems. 2018 Oct 9;3(5):e00137-18. doi: 10.1128/mSystems.00137-18. PMID: 30320222; PMCID: PMC6178585.
Response: Thank you very much for your valuable comments. Fecal microbial transplantation is indeed a very popular research area at the moment, and we have cited this excellent research in the section "2.3. Intestinal microbiota" on page 3 of the manuscript.
Reviewer 4 Report
The paper is very interesting and it opens new therapeutical solutions based on Traditional Chinese Medicine. I suggest:
- At page 4, line 150, the authors should precise the type of bacteria that alter intestinal microflora
- At page 10, line 436, the authors should also mention the role of Zinc-Carmosin in the management of radiation colitis and other inflammatory bowel disease
- At page 15, line 626, the authorss should insert a table illustrating the indication of Chinese medication in different intestinal pathologies
Author Response
Responses to the reviewer4 comments
1.At page 4, line 150, the authors should precise the type of bacteria that alter intestinal microflora
Response: Thank you for your valuable comments. We are not sure that we really understood what you meant and we have made changes in the manuscript as we understand them. If the changes are not correct, we would like to ask you to make them again and thank you again for scrutinizing the manuscript.
2.At page 10, line 436, the authors should also mention the role of Zinc-Carmosin in the management of radiation colitis and other inflammatory bowel disease
Response: Research related to zinc-carnosine for the treatment of intestinal diseases is currently very hot. There is a real need for additional clarification in this area in our manuscript. We have added some of the zinc-carnosine research results to the "4.1. PI3K/Akt/mTOR signaling pathway" section of the manuscript. Thank you again for your valuable comments on the manuscript.
3.At page 15, line 626, the authorss should insert a table illustrating the indication of Chinese medication in different intestinal pathologies
Response: Thanks for your great suggestion. We have modified Table 1, which covers the indications of Chinese medication for different intestinal pathologies. Thanks again for your careful review of the manuscript.
Round 2
Reviewer 2 Report
The authors have made a reasonable effort in addressing reviewer's comments. Proofreading is suggested since there are some errors/typos in the manuscript including the first word on the title (should read: "Critical").
English is OK proofreading for typos is suggested.
Author Response
Responses to the reviewer2 comments
1.The authors have made a reasonable effort in addressing reviewer's comments. Proofreading is suggested since there are some errors/typos in the manuscript including the first word on the title (should read: "Critical")
Response: We are very sorry that due to our carelessness, the first word of the manuscript was misspelled. We have rechecked and corrected any errors in the manuscript, and thank you very much for your careful review.